# Granulins Regulate Aging Kinetics in the Adult Zebrafish Telencephalon

**DOI:** 10.3390/cells9020350

**Published:** 2020-02-03

**Authors:** Alessandro Zambusi, Özge Pelin Burhan, Rossella Di Giaimo, Bettina Schmid, Jovica Ninkovic

**Affiliations:** 1Institute of Stem Cell Research, Helmholtz Center Munich, 85764 Neuherberg, Germany; 2Graduate School of Systemic Neuroscience; Biomedical Center, Faculty of Medicine, LMU Munich, 82152 Planegg, Germany; 3German Center for Neurodegenerative Diseases (DZNE), 81377 München, Germany; oezge-pelin.burhan@dzne.de (Ö.P.B.); bettina.schmid@dzne.de (B.S.); 4Department of Biology, University of Naples Federico II, 80134 Naples, Italy; digiaimo@unina.it

**Keywords:** neurogenesis, microglia, granulin, aging

## Abstract

Granulins (GRN) are secreted factors that promote neuronal survival and regulate inflammation in various pathological conditions. However, their roles in physiological conditions in the brain remain poorly understood. To address this knowledge gap, we analysed the telencephalon in Grn-deficient zebrafish and identified morphological and transcriptional changes in microglial cells, indicative of a pro-inflammatory phenotype in the absence of any insult. Unexpectedly, activated mutant microglia shared part of their transcriptional signature with aged human microglia. Furthermore, transcriptome profiles of the entire telencephali isolated from young Grn-deficient animals showed remarkable similarities with the profiles of the telencephali isolated from aged wildtype animals. Additionally, 50% of differentially regulated genes during aging were regulated in the telencephalon of young Grn-deficient animals compared to their wildtype littermates. Importantly, the telencephalon transcriptome in young Grn-deficent animals changed only mildly with aging, further suggesting premature aging of Grn-deficient brain. Indeed, Grn loss led to decreased neurogenesis and oligodendrogenesis, and to shortening of telomeres at young ages, to an extent comparable to that observed during aging. Altogether, our data demonstrate a role of Grn in regulating aging kinetics in the zebrafish telencephalon, thus providing a valuable tool for the development of new therapeutic approaches to treat age-associated pathologies.

## 1. Introduction

Aging is associated with numerous changes in a wide range of processes, thus progressively leading to impaired organ function and increasing the risk of the onset of numerous pathologies. Accordingly, brain aging is one of the greatest risk factors for multifactorial diseases including Alzheimer’s and Parkinson’s diseases [1]. Typical aging hallmarks such as genomic instability, telomere shortening, metabolic dysregulation, cellular senescence, stem cell exhaustion, and decreased neurogenesis are observed in the brain and affect a broad range of cellular functions [2], thereby leading to impaired tissue homeostasis and regeneration. A primary reason for this impaired tissue homeostasis and regeneration is age-related exhaustion of tissue-specific stem cells, a process observed in various organs [3,4,5], including the mammalian brain [6,7,8,9,10]. Although prominent age-associated changes, including decreased neurogenesis and changes in the lateral ventricle choroid plexus [11,12,13], have been detected in the aging mammalian brain, comparisons of neural stem cell transcriptomes have revealed a remarkably small set of differentially regulated transcripts, which are largely associated with cell cycle regulation, neuronal differentiation, and inflammation [10,14,15,16]. These analyses support the hypothesis that specific changes in a limited number of key regulators influenced by both intrinsic [16] and extrinsic [11,12,17,18] aging-associated pathways lead to the aging of the neural stem cell compartment. Interestingly, a prominent phenomenon associated with aging in the neural stem cell niche is the activation of resident microglial cells, which develop a pro-inflammatory phenotype [19,20,21,22]. This process results in the activation of NF-kB signalling and enrichment in pro-inflammatory cytokines, with increased production of IL-1b, TNF, and interferons- typical activators of pro-inflammatory responses [23]. Indeed, the activation of NF-kB signalling in the hypothalamus decreases the production of gonadotropin-releasing hormone in neurons [24], thus causing various age-associated changes in stem cell compartments of the brain, but also modulates systemic aging, including muscle weakness and skin atrophy [24]. Furthermore, the mRNA decay factor AUF1 regulates both the aging-associated inflammatory response and maintenance of telomere length [25], thereby suggesting not only that the interactions among different cells in the stem cell compartment contributes to tissue aging but also that the same factor might be involved in both cell-intrinsic and -extrinsic pathways involved in the aging process. Therefore, one major challenge is to better understand the relationship among these processes in different cell types and to identify novel pathways and regulators contributing to aging regulation. These pathways would be suitable targets for the development of new therapeutic approaches aiming to limit or block the detrimental phenotypes acquired with aging, thus decreasing the risk of associated pathologies.

Progranulin (GRN) is a secreted factor that is mainly expressed in microglia and neurons, and can be proteolytically processed into granulin peptides [26,27,28]. Both GRN and granulin peptides are biologically active [29,30]. Granulin deficiency shortens lifespan in mice [31] and African turquoise killifish [32], in line with the association of genetic variants in the *GRN* gene with age-associated phenotypes in the human cerebral cortex [33]. Moreover, mutations in the human *GRN* gene are associated with early onset of age-related neurodegenerative diseases, such as frontotemporal lobar degeneration (FTLD) [34,35] and neuronal ceroid lipofuscinosis (NCL) [36,37]. Additionally, GRN regulates aging-related processes such as inflammation [38,39,40,41,42,43] and neuronal survival [44,45,46], thus supporting a possible role of GRN in the regulation of brain aging. 

Although GRN is associated with aging in the human cerebral cortex, the specific molecular mechanisms and cellular basis leading to the accelerated aging phenotypes remain poorly understood, possibly because of the opposing roles of GRN and granulin peptides, generated by GRN proteolysis in mammals [29,30]. Whereas GRN, for example, has an anti-inflammatory function, some granulin peptides enhance the inflammatory response [29,30]. In contrast to mammals, zebrafish have two orthologs of *Grn*, *granulin a* (*grna*) and *granulin b* (*grnb*), in addition to two paralogs, *granulin 1* (*grn1*) and *granulin 2* (*grn2*), which encode shorter proteins similar to the products of mammalian GRN proteolytic cleavage [47]. Therefore, the zebrafish model of aging [48,49,50], combined with a recently generated double mutant for Grna and Grnb, offers the possibility of determining the specific functions of GRN and granulin peptides in brain aging [51].

## 2. Materials and Methods

### 2.1. Animal Experiments

Adult (3–5-month-old) and aged (15–24-month-old) *grna^+/+^;grnb^+/+^* and *grna^−/−^;grnb^−/−^* zebrafish siblings [51], and zebrafish from the AB/EK strain and from the transgenic lines *Tg* (*olig2:DsRed*) [52], *Tg* (*mpeg1:mCherry*) [53], *Tg* (*olig2:DsRed;grna^−/−^;grnb^−/−^*) and *Tg* (*mpeg1:mCherry;grna^−/−^;grnb^−/−^*) were used in all experiments. The fish were reared under standard husbandry conditions [54], and experiments were performed according to the handling guidelines and regulation of EU and the Government of Upper Bavaria (AZ 55.2-1-54-2532-09-16).

### 2.2. DNA Extraction and Genotyping

Genotyping of *grna* and *grnb* was performed by cutting a small part of the zebrafish tail fin. REDExtract-N-Amp Tissue Kit (Sigma-Aldrich^®^, Merck, Darmstadt, Germany) was used to rapidly extract the genomic DNA from animal tissues according to the manufacturer´s instructions. Isolated genomic DNA was amplified by PCR with the following primers: *grna* forward (TTCAGTCATTGTTTCAGAGGTCA), *grna* reverse (TTCCTCTGATCCACTTTCTACCA), *grnb* forward (AATGACACAAGACGTCCTCATAAA) and *grnb* reverse (AAAAATAATAACCACAGCGCAACT). Sanger sequencing was then performed to obtain and compare the nucleotide sequences of different samples and identify the selected mutations for *grna* and *grnb*. The *grna* selected mutation was identified by a deletion of 11 nucleotides, which caused a frameshift of the open reading frame and resulted in a premature STOP codon. The *grnb* selected mutation was identified by a deletion of 13 nucleotides and insertion of 9 nucleotides, which caused a frameshift of the open reading frame and resulted in a premature STOP codon.

### 2.3. Tissue Preparation and Immunohistochemistry

Animals were sacrificed through overdose of tricaine methane sulfonate (MS222, 0.2%) via prolonged immersion. Tissue processing was performed as previously described [55]. For cell immunolabelling, we used rat anti-BrdU (1:200, ab6326, Abcam, Berlin, Germany), mouse anti-fish leukocytes 4C4 antigen (1:500, 92092321, 7.4.C4, Health Protection Agency Culture Collections, Salisbury, England), rabbit anti-HuC/HuD (1:500, 210554, Abcam, Berlin, Germany) and rabbit anti-Sox10 (1:2000, GTX128374, Biozol, Eching, Germany). The primary antibodies were detected with specific secondary antibodies labelled with Alexa 488, Alexa 546, and Alexa 633 (1:1000, Invitrogen, Thermo Fisher, Dreieich, Germany). Sections were embedded in Aqua Polymount (Polyscience, Hirschberg an der Bergstraße, Germany). 

Immunodetection of BrdU required a pretreatment with 4 N HCl followed by washes with borate buffer and PBS before the sections were immersed in the anti-BrdU antibody. All antibodies were dissolved in 0.5% Triton X-100 and 10% normal goat serum.

### 2.4. Bromodeoxyuridine (BrdU) Labelling Experiments

To analyse the proliferative capacity of glial cells, we performed long-term bromodeoxyuridine (BrdU) (Sigma Aldrich) incorporation. Fish were kept in BrdU-containing aerated water (10 mM) for 16 h/day during 5 consecutive days for long-term analysis. This time frame was determined to label a substantial proportion of activated neural stem cells that would generate neurons, based on the basis of the observations that adult neural stem cells (aNSCs) are largely quiescent in the intact adult zebrafish telencephalon [56]. During the 8 h outside BrdU-containing water, fish were kept in fresh water and fed. Animals were sacrificed 14 days after BrdU treatment (5 days BrdU water + 14 day chase), because this time point was previously described to have the greatest number of newly generated neurons incorporating BrdU in the adult zebrafish telencephalon [57].

### 2.5. Image Acquisition and Processing

All images were acquired with an Olympus FV1000 cLSM system (Olympus, Tokyo, Japan), using the FW10-ASW 4.0 software (Olympus, Tokyo, Japan). Quantifications and co-localisation analysis were performed in Imaris software version 8.4 (Bitplane, Zurich, Switzerland) and ImageJ software (National Institutes of Health, Bethesda, MD, USA). 

### 2.6. Quantitative Analysis and Statistical Tests

For all analyses, the group size was previously determined to have a statistical power ≥0.95, identifying a group size of at least four independent telencephali. The number of telencephali analysed is specified in the figure legends and in the graphs, where each data point represents a distinct biological replicate. 

All the sections belonging to the telencephalon were quantified (sections containing the olfactory bulb or optic tectum were excluded). All quantifications are displayed as cell density, defined as the total number of cells occupying a specific area (ventricular or parenchymal zone). The volumes (mm^3^) of either ventricular (Figure 6c) or parenchymal (Figure 7b) zones were calculated in Imaris software version 8.4 (Bitplane, Zurich, Switzerland). For normalisation, the total numbers of cells occupying a specific zone were divided by the calculated volumes of the corresponding zones.

Data are presented as the mean with SEM, and each data point represents a distinct biological replicate. Statistical analysis was performed with one-way ANOVA test for the oligodendrogenesis experiment (Figure 7) and two-way ANOVA test for the microglial morphology experiment (Figure 2), the constitutive neurogenesis experiment (Figure 6) and the comparison of relative telomere length (Figure 9b). GraphPad Prism Version 8.2.0 was used for the statistical tests. Statistical analysis for data associated with Figure 4a, Figure 9a, and Appendix A was automatically performed with the DESeq2 pipeline (release 3.9) for differential expression analysis. 

### 2.7. Morphological Analysis of Microglial Cells

To assess and compare the morphology of microglial cells between different groups in Figure 2, we quantified the average process length (μm), number of main processes and area of microglial somata (μm^2^) in ImageJ (Fiji) software. A total of 80 cells from four different telencephali per group were analysed.

### 2.8. Fluorescence-Activated Cell Sorting (FACS) Isolation Method

Animals from the *Tg* (*olig2:DsRed*), *Tg* (*mpeg1:mCherry*), *Tg* (*olig2:DsRed; grna^−/−^;grnb^−/−^*), and *Tg* (*mpeg1:mCherry; grna^−/−^;grnb^−/−^*) transgenic lines were sacrificed through MS222 overdoses, the telencephalon was dissected from each single animal, and three telencephali were pooled for each replicate. A single-cell suspension was prepared as previously described [58], and cells were analysed with a BD FACSAria III instrument in BD FACS Flow TM medium. Debris and aggregated cells were gated by forward scatter-sideward scatter; single cells were gated in by FSC-W/FSC-A. Gating for fluorophores was performed using AB/EK animals as a baseline (Appendix A). Cells were directly sorted into extraction buffer from PicoPure RNA isolation kit (Thermo Fisher, Dreieich, Germany).

### 2.9. Libraries for Deep Sequencing of Whole Telencephali and FACS Sorted Cells 

Total RNA from the entire telencephalon was isolated with Qiagen RNeasy kit, whereas total RNA from sorted cells was isolated with PicoPure RNA isolation kit from Thermo Scientific, according to manufacturer’s instructions. 

The quality and concentration of RNA were assessed on an Agilent 2100 Bioanalyzer. Only RNA with a RIN value ≥ 8 were used for library preparation. cDNA was synthesised from 10 ng of total RNA (whole telencephalon) or 500–1000 pg of total RNA (fluorescence activated cell sorting (FACS) isolated cells) with a SMART-Seq v4 Ultra Low Input RNA kit for Sequencing (Takara, Mountain View, CA, USA), according to the manufacturer’s instructions. cDNA quality and concentration were assessed on an Agilent 2100 Bioanalyzer before library preparation with MicroPlex Library Preparation kit v2 (Diagenode, Liège, Belgium). All libraries (five replicates per condition in the case of samples from the whole telencephalon and three or four replicates per condition in the case of samples from FACS isolated cells) were processed together to minimise batch effects. Final libraries were evaluated and quantified with an Agilent 2100 Bioanalyzer, and the concentration was measured additionally with a Qubit 4 instrument and the Qubit dsDNA HS assay (Thermo Fisher, Dreieich, Germany) before sequencing. The uniquely barcoded libraries were multiplexed onto one lane, and 100-bp paired-end deep sequencing was carried out on a HiSeq 4000 instrument (Illumina, San Diego, CA, USA), generating approximately 20 million reads per sample. FASTQ files are available at GEO database with accession number GSE144543. FASTQ files were mapped with the STAR pipeline to the annotated zebrafish genome (GRCz10) [59]. DESeq2 (release 3.9) for R environment was used for differential expression analysis based on raw counts [60]. The rlog transformation method was applied for visualisation (heat maps). The cutoff for the differentially regulated genes was based on the fold change expression (twofold or greater) and the adjusted *p* value equivalent to the 5% false discovery rate (*padj* ≤ 0.05), according to the DESeq2 pipeline [60]. Gene ontology (GO) enrichment analysis was performed with DAVID bioinformatics resources 6.8 (*p* ≤ 0.05, fold enrichment ≥ 1.5) [61,62], and the relevant enriched pathways were manually selected. 

### 2.10. Quantification of Relative Telomere Length

For qPCR telomere length measurements, genomic DNA was obtained from whole zebrafish brain extracted from young (3–5-month-old) and aged (15–24-month-old) adult *grna^+/+^;grnb^+/+^* and *grna^−/−^;grnb^−/−^* zebrafish siblings with QIAamp DNA Mini Kit (Qiagen, Hilden, Germany). Telomere primer sequences were obtained from a previous publication [63]: *dio2* forward: GGGGACGGCAGAAGAAATGA, *dio2* reverse: AGGTCCACACTAAGCAAGCC, *telo* forward: CGGTTTGTTTGGGTTTGGGTTTGGGTTTGGGTTTGGGTT and *telo* reverse: GGCTTGCCTTACCCTTACCCTTACCCTTACCCTTACCCT. A single-copy gene, iodothyronine type II (*dio2*), was used as a reference gene for the quantification. The ratio of amplified telomere signal to single-copy gene signal was calculated as previously described to obtain the relative telomere length [64].

## 3. Results

### 3.1. Grna and Grnb Deficiency Leads to a Pro-Inflammatory Transcriptional Signature in Microglial Cells in the Adult Zebrafish Telencephalon

Granulin is a key factor mainly expressed in microglia and neurons that is involved in the regulation of several processes including inflammation in the mouse cerebral cortex [38,39]. For this reason, we decided to study the role of the zebrafish structural orthologs of mouse GRN, Grna and Grnb, in the context of the immune system in the adult zebrafish telencephalon. To this end, we used FACS to isolate the resident brain immune cells, microglia, from the intact telencephali of *Tg* (*mpeg1:mCherry*) (further referred to as wildtype) and *Tg* (*mpeg1:mCherry;grna^−/−^,grnb^−/−^)* (further referred to as mutant) animals, on the basis of the mCherry expression in Mpeg1-positive microglia (Figure 1a–d; Appendix A). Importantly, the identity of mCherry^+^ cells was validated by co-staining with previously reported microglial markers, such as 4C4 (Figure 1b–c′). 

To determine the purity of isolated cells, we analysed the expression of typical microglial, neuronal, and oligodendroglial genes [65] in the sorted mCherry^+^ population (Appendix A). Significant enrichment in microglial genes was detected in Mpeg1^+^ isolated cells, thus confirming the purity of our enrichment (Appendix A). Importantly, *grna* was strongly enriched in Mpeg1^+^ cells, in line with high *Grn* expression in mammalian microglia [27], whereas *grnb* was expressed at lower levels (Appendix A). 

The comparison of the transcriptomes of wildtype and mutant Mpeg1^+^ cells and differential expression analysis (DEseq2) revealed 1567 significantly upregulated and 2018 significantly downregulated genes in mutant Mpeg1^+^ cells (*padj* ≤ 0.05; 1 ≤ log2FC ≤ −1) (Figure 1e; Appendix A). GO analysis (based on DAVID 6.8) revealed an enrichment in genes associated with inflammation, apoptosis, cell proliferation and extracellular matrix composition (Figure 1f,g; Appendix A). Notably, many upregulated genes in mutant Mpeg1^+^ cells were associated with the tumor necrosis factor receptor, the MAPK and the JAK/STAT signalling pathways, which are involved in the induction of pro-inflammatory cytokine expression [66,67] (Figure 1f and Appendix A). Furthermore, we identified numerous downregulated genes belonging to the PPAR signalling pathway, which has a role in counteracting the expression of pro-inflammatory cytokines [68] (Figure 1g and Appendix A). These results indicate a switch toward a pro-inflammatory state of microglial cells in the adult telencephalon in Grna; Grnb-deficient animals. To confirm this hypothesis, we analysed the expression levels of several cytokines associated with pro-inflammatory and anti-inflammatory microglial phenotypes and detected higher mRNA levels of pro-inflammatory cytokines including *il6*, *tnfa*, *il12bb*, and *Cxcl8a* (*il8*) in mutant Mpeg1^+^ cells (Appendix A; Appendix A). In contrast, the mRNA levels of anti-inflammatory cytokines including *tgfb3*, *igf1*, and *il10* were significantly lower in mutant Mpeg1^+^ cells than wildtype Mpeg1^+^ cells (Appendix A; Appendix A), thus demonstrating their activated state. The numerous transcriptional changes observed in mutant Mpeg1^+^ cells contrast with the results obtained from GRN-deficient microglia in the mouse brain [38]. In contrast to our observations, GRN-deficient microglia in the mouse brain display different transcriptional profiles only during advanced aging [38], thereby suggesting that granulins might regulate microglia activation with different kinetics in the mouse and zebrafish brains. We compared the differentially expressed genes (DEGs) in our dataset with age-dependent upregulated genes in the cerebral cortex in GRN-deficient mice enriched in two major categories (immune response and lysosomal pathway) that showed exclusive association with the microglial population in mice (Appendix A) [38]. Interestingly, some of the genes that displayed age-dependent upregulation in the cerebral cortex in GRN-deficient mice were also upregulated in young mutant Mpeg1^+^ cells (Appendix A), thus indeed suggesting that granulin age-dependent role in suppressing aberrant microglial activation might be fundamental already during young adulthood in the adult zebrafish brain.

Because the transcriptome analysis suggested Grna; Grnb-dependent changes in the activation state of microglial cells, we further examined their morphology as an activation hallmark. We labelled microglial cells in the intact adult zebrafish telencephalon through immunohistochemistry against 4C4 [69] and compared 4C4^+^ cells in the telencephalon in young wildtype (3–5-month-old) and mutant (3-5-month-old) animals (Figure 2a–b″,e–g). Wildtype 4C4^+^ cells displayed a typical elongated morphology, characteristic of resident non-activated microglia (Figure 2a–a″,e–g). In contrast, most mutant 4C4^+^ cells displayed larger somata, and hyper-ramified and shorter processes-morphological characteristics typical of the activated state (Figure 2b–b″,e–g). 

Together, our transcriptome and morphological analyses demonstrate the roles of Grna and Grnb in tuning the activation state of microglial cells in the zebrafish telencephalon. Grna and Grnb deficiency promotes the switch toward a pro-inflammatory and possibly detrimental microglial phenotype.

### 3.2. Loss of Grna; Grnb Function Induces Premature Microglial Aging

The pro-inflammatory phenotype of microglial cells, including both morphological and characteristic transcriptomic changes, has been associated with aging [70,71]. Indeed, most 4C4^+^ cells in old (15–24-month-old) wildtype animals displayed larger somata and shorter processes (Figure 2c–c″,e–g), in agreement with findings from previous reports [70,71]. For these reasons, we speculated that Grna; Grnb deficiency might cause a premature aging phenotype in microglial cells in the telencephalon in young animals. Therefore, we analysed and compared the morphology of 4C4^+^ cells in young (3–5-month-old) mutant and old (15–24-month-old) wildtype animals (Figure 2b–c″,e–g). Notably, we observed similar 4C4^+^ cell morphology in the telencephali of young mutant and old wildtype animals (Figure 2b–c″,e–g). Unexpectedly, we detected no additional morphological changes in 4C4^+^ cells of old (15–24-month-old) mutant animals (Figure 2d–d″,e–g). Together, our results indicate a premature aging phenotype in Grna; Grnb-deficient microglia.

To substantiate this hypothesis, we compared the DEGs in young Grna; Grnb-deficient Mpeg1^+^ cells with the dataset of DEGs of aged (~95-years-old) human microglial cells [70] (Figure 3). Interestingly, 271 DEGs identified in young Grna; Grnb-deficient Mpeg1^+^ cells showed the same regulation pattern seen in aging human microglia (~10% of all human DEGs with existing zebrafish orthologs) (Figure 3a). Notably, GO analysis of overlapping genes revealed an enrichment in genes associated with cholesterol biosynthesis, aging, cell cycle, multiple sclerosis, and Alzheimer’s disease (Figure 3b and Appendix A).

In summary, our results support roles of Grna and Grnb in regulating the aging kinetics of microglial cells. Furthermore, these results are in agreement with the described role of GRN in suppressing excessive microglial activation [38] and with previous studies identifying the GRN gene locus as a determinant of accelerated aging in the human cerebral cortex and short lifespan in the African turquoise killifish [32,33].

### 3.3. Transcriptome Analysis Reveals Premature Aging of Grna; Grnb-Deficient Brains

To assess whether Grna and Grnb deficiency might cause a premature aging phenotype exclusively in microglial cells, or affected other cell types and processes in the zebrafish telencephalon, we compared the total mRNA from whole telencephali of young wildtype (3–5-month-old), old wildtype (15–24-month-old), young mutant (3–5-month-old), and old mutant (15–24-month-old) animals (Figure 4; Figure 5).

First, we analysed Grna and Grnb mRNA levels in young and old wildtype animals. In contrast to the unchanged Grna mRNA levels, the Grnb mRNA levels were significantly lower in old wildtype animals (Figure 4a). For this reason, we speculate that Grnb deficiency at the stage of young adulthood (3–5-month-old) might mimic the physiological decrease observed in old animals, thus accelerating aging kinetics in the brain.

The differential expression analysis between young and old wildtype telencephali revealed numerous transcriptional changes associated with aging in old wildtype animals, in which we identified 1207 significantly upregulated and 1357 significantly downregulated genes (*padj* ≤ 0.05; 1 ≤ log2FC ≤ −1) (Figure 4b; Appendix A). Similarly, when we compared the transcriptomes of young mutant and young wildtype animals, we identified 1142 significantly upregulated and 941 significantly downregulated genes (*padj* ≤ 0.05; 1 ≤ log2FC ≤ −1) (Figure 4c; Appendix A). Additionally, in a direct comparison of the transcriptomes of old wildtype and young mutant animals, we detected fewer DEGs (Figure 4d; Appendix A). Notably, the comparison of the transcriptomes of young mutant and old mutant animals indicated only 178 significantly upregulated and 241 significantly downregulated genes (Figure 4e; Appendix A), thus supporting the shift in the brain-wide transcriptome in young mutant animals toward that of aged brains. 

We further selected the top 100 significantly upregulated and top 100 significantly downregulated genes in old wildtype animals (using young wildtype animals as a reference) and compared the rlog-transformed values representing the expression levels of these genes in different ages and genotypes (Figure 5a,b). Unexpectedly, most selected genes had comparable rlog-transformed values in old wildtype, young mutant, and old mutant telencephali (Figure 5a,b). These data provide additional evidence of premature aging in the brain in young Grna; Grnb-deficient animals.

Our transcriptome analysis of microglia, however, highlighted numerous transcriptomic changes in mutant microglial cells, thus indicating a possibility that the transcriptional changes in mutant microglia might largely contribute to the transcriptional changes observed in the entire telencephalon. To address this possibility, we analysed the expression of the top 100 upregulated and top 100 downregulated genes, as described above, in wildtype Mpeg1^+^ and mutant Mpeg1^+^ cells. Indeed, only a small proportion of selected genes were also regulated with the same pattern in mutant microglial cells (Figure 5a,b). To further address the total contribution of microglia-specific transcriptomic changes to the overall changes in the whole telencephalon in young mutant animals, we compared the DEGs in mutant Mpeg1^+^ cells and in the whole telencephalon in young mutant animals and we detected an overlap of 10.9%. Together, our data demonstrate that Grna and Grnb deficiency promote additional transcriptional changes beyond those specifically detected in microglial cells. 

To specifically identify the size and nature of commonly DEGs in old wildtype and young mutant telencephali, we selected and compared the common upregulated and downregulated genes in old wildtype and young mutant animals (using young wildtype animals as a reference) (Figure 5c,d). Notably, 50.7% of upregulated genes and 48.8% downregulated genes were shared between old wildtype and young mutant animals (Figure 5c,d). GO analysis revealed common enrichment in genes associated with programs previously identified in the transcriptome of *Notobranchius furzeri*, a well-established model of aging, including regulation of apoptosis, immune response, extracellular matrix composition and cell adhesion [72] (Figure 5e,f and Appendix A). 

Because GRN has been found to be involved in lysosomal homeostasis and GRN-deficient mice displayed lysosomal dysfunction, we analysed mRNA expression levels of genes associated with lysosomal regulation differentially regulated in GRN-deficient mice [73,74]. In contrast to the results in GRN-deficient mice, expression of lysosomal genes was only slightly changed in the adult telencephalon in young Grna; Grnb-deficient zebrafish, as previously described [51], and was not further changed in old mutant brains, thereby suggesting a discrepancy between the lysosomal regulatory function of granulins in mouse and zebrafish brains (Appendix A). The observed discrepancy may be explained by the possible involvement of *grn1* and *grn2* in regulating the lysosomal homeostasis in the adult zebrafish brain, thus compensating for the loss of Grna and Grnb.

Collectively, our data indicate that aging causes numerous transcriptional changes in the brain in old wildtype animals. A significant proportion of these changes was mimicked by Grna; Grnb deficiency in the brain in young animals and was not limited to microglial cells. Furthermore, in Grna; Grnb-deficient animals, these changes are significantly reduced between young and old animals, thus strengthening our hypothesis that Grna and Grnb deficiency causes a premature aging phenotype in the brain in young animals.

### 3.4. Grna and Grnb Deficiency Causes a Reduction in Neurogenesis in the Ventricular Zone of the Zebrafish Telencephalon, thus Mimicking the Phenotype of Reduction Observed during Aging 

To understand the aging-related biological processes affected by Grna; Grnb-deficiency, we further analysed shared DEGs in old wildtype and young mutant animals. We identified a significant downregulation of *il4* and *stat6* (Figure 6a). These genes have been implicated in the regulation of age-related inflammatory changes in the rat hippocampus and have a pivotal role in the activation of restorative neurogenesis in a zebrafish model for Alzheimer’s disease [75,76]. 

Specifically, in response to the injection of Aβ42 derivatives, Il4 is activated and promotes stem cell proliferation and neurogenesis via Stat6 phosphorylation [75]. Furthermore, neurogenesis has been demonstrated to decline with aging: ependymoglial cells (neural stem cells in the adult zebrafish telencephalon) enter the cell cycle less frequently, thus producing fewer neuroblasts and consequently fewer neurons [77]. For these reasons, we hypothesised that Grna and Grnb deficiency in young mutant animals might affect neurogenesis as a hallmark of aging, and the downregulation of *il4* and *stat6* might be a possible underlying mechanism. To verify this hypothesis, we assessed the proliferative capacity and the neurogenic potential of young wildtype, old wildtype, young mutant, and old mutant animals in the ventricular zone (VZ) of the zebrafish telencephalon (Figure 6b–l). The VZ is a neurogenic niche that hosts somata of ependymoglial cells, which produce new neurons throughout the lifetime [56,57], but show an age-dependent decline [77]. Newborn neurons are generated and deposited in the area immediately below the VZ, the subventricular zone (SVZ) [78]. In this analysis, animals were kept in BrdU water for 5 consecutive days to label all stem cells and neuronal progenitors going through S-phase during the labelling phase (Figure 6b). After the labelling period, we kept animals in normal water for 2 weeks to allow neuronal differentiation of cells that exited the cell cycle and dilution of the label in the still-cycling progenitor population. Therefore, the Brdu^+^ HuCD^−^ cells were classical label-retaining neural stem cells [13]. The number of newborn neurons was assessed by quantification of BrdU^+^ HuCD^+^ (marker for post-mitotic neurons) cells in the VZ and SVZ (25 μm from the outer layer of the ventricle, as indicated in Figure 6c) (Figure 6d–g″,h). In agreement with previously published data, we detected a significant decrease in the number of BrdU^+^ HuCD^+^ cells (newborn neurons) in the VZ in old wildtype animals (Figure 6d–e″,h). To assess whether Grna and Grnb deficiency might affect the constitutive neurogenic potential to the same extent as that observed in old wildtype animals, we quantified the number of BrdU^+^ HuCD^+^ in young mutant animals (Figure 6d–f″,h) and we detected significantly fewer newborn neurons (BrdU^+^ HuCD^+^) in the VZ in young mutant animals (Figure 6f–f″,h). Interestingly, the numbers were comparable to those observed in old wildtype animals and did not further decrease in old mutant animals, in agreement with the absence of further transcriptional changes between young and old mutant animals (Figure 6g–g″,h). Together, our results demonstrate that aging causes a significant decline in constitutive neurogenesis in the zebrafish telencephalon, and that the phenotype of this decrease is mimicked by Grna and Grnb deficiency in the telencephalon in young animals. This analysis further supports Grna and Grnb being key regulators of aging kinetics in the zebrafish brain. Moreover, we detected a ~50% reduction in the number of BrdU^+^ (label retaining) cells in the VZ in young mutant animals (Figure 6d,f,i). Interestingly, the number of BrdU^+^ cells in the VZ in young mutant animals was comparable to the number of BrdU^+^ cells observed in the VZ in old wildtype animals (Figure 6d–f,i). Notably, no further decrease in the number of BrdU^+^ cells was observed in the VZ in old mutant animals (Figure 6d–g,i). 

Because ependymoglial cells enter the cell cycle less frequently during aging [77], we speculated that the decreased constitutive neurogenesis in old wildtype and both young and old mutant animals might be explained by an increased number of quiescent ependymoglial cells that do not enter the cell cycle and therefore do not incorporate BrdU. Therefore, we analysed the number of BrdU^+^ HuCD^−^ cells in the neurogenic niche (Figure 6l). Notably, we detected a significant decrease in the number of BrdU^+^ HuCD^−^ cells in old wildtype animals, to a level comparable to the number of BrdU^+^ HuCD^−^ cells in young and old mutant animals (Figure 6l). Therefore, these data suggest the premature entry to quiescence of neural stem cells in Grna; Grnb-deficient animals and demonstrate a clear decline in adult neurogenesis, as hallmark of aging. Finally, we quantified the cell density of total HuCD^+^ neurons in the parenchymal area among different groups and observed no significant differences (Figure 6m), a finding in line with different regulatory mechanisms underlaying embryonic and adult neurogenesis. 

### 3.5. Grna and Grnb Deficiency Leads to Impaired Oligodendrocyte Precursor Cells Differentiation

Along with a decline in neurogenesis, aging causes a decline in oligodendrogenesis, thus decreasing the abundance of oligodendrocyte precursor cells (OPCs) in the parenchymal area of the adult zebrafish telencephalon [77]. To verify this concept, we kept animals in BrdU water for 5 consecutive days to label OPCs and performed a 14-day chase (Figure 7a). Oligodendrocyte lineage cells that proliferated during the labelling time were identified by double labelling for BrdU and Sox10 (an oligodendrocyte lineage marker) in the parenchymal area (Figure 7b). In old wildtype animals, we detected a threefold decrease in the number of BrdU^+^ Sox10^+^ cells (Figure 7c–d″,f) in line with findings from previous reports [77]. Unexpectedly, young mutant animals showed a similar decrease (Figure 7e–e″,f). Together, our data indicate that both aging and Grna; Grnb deficiency induce a decrease in oligodendrogenesis in the zebrafish brain parenchyma. 

The observed phenotype could potentially be explained by lower proliferation of OPCs or impaired oligodendrogenesis. To distinguish between these two possibilities, we assessed the transcriptomic changes in oligodendroglial cells from Grna; Grnb-deficient animals. By using FACS, we isolated oligodendroglial cells from the *Tg* (*olig2:DsRed*) animals according to the DsRed expression in Olig2-positive oligodendroglial cells (Figure 8a–c and Appendix A). Importantly, the identity of cells was validated by co-staining with previously reported oligodendroglial marker, such as Sox10 (Figure 8a–b′) [79]. 

To determine the purity of isolated cells, we analysed the expression of typical oligodendroglial, neuronal and microglial genes [65] in the sorted DsRed^+^ population (Appendix A). Significant enrichment in oligodendroglial genes was detected in Olig2^+^ isolated cells together with expression of neuronal genes, in line with the expression of Olig2 in a small subset of neuronal progenitors in the ventral telencephalon (Appendix A) [79]. Interestingly, we observed comparable expression levels of *grnb* in Mpeg1^+^ and Olig2^+^ cells (Appendix A), thus suggesting a possible previously undescribed direct role of granulins in oligodendrocyte cells.

By comparing the transcriptomes of young wildtype and mutant Olig2^+^ cells, we detected 2835 significantly upregulated and 3300 significantly downregulated genes in mutant Olig2^+^ cells (*padj* ≤ 0.05; 1 ≤ log2FC ≤ −1) (Figure 8d; Appendix A). GO analysis of significantly upregulated genes in mutant Olig2^+^ cells revealed overrepresentation of terms associated with the immune response, chemokine activity and cell chemotaxis (Figure 8e). Notably, GO analysis of significantly downregulated genes in mutant Olig2^+^ cells revealed enrichment in several terms associated with pathways involved in oligodendroglial cell differentiation and myelination, including negative regulation of BMP, Hedgehog, Notch, TGF-β and ErbB signalling pathways [80,81,82,83,84] (Figure 8f; Appendix A). These results, together with the observed decrease in number of BrdU^+^ Sox10^+^ cells in the telencephalon in young mutant animals, strongly suggest that Grna and Grnb deficiency affects the normal differentiation process of OPCs. Furthermore, these data suggest that the decreased oligodendrogenesis observed during aging might be partially explained by the decreased granulin levels.

### 3.6. Grna and Grnb Deficiency Decreases of Telomere Length 

Grna; Grnb deficiency leads to numerous aging-associated changes in the adult zebrafish brain. Therefore, we investigated telomere length, an established cell-intrinsic feature of aging cells. Telomeres are known to shorten with aging [2,48,85]. First, we addressed the expression levels of genes involved in telomere protection (Figure 9a). Indeed, we detected a significant downregulation of genes preventing telomere shortening (*tert*, *tp53*, and *tpp1*) in old wildtype, young mutant, and old mutant animals (Figure 9a). To examine the relevance of these genes to telomere length regulation, we quantified the relative telomere length in the brain in young wildtype, old wildtype, young mutant, and old mutant animals, as previously described [64]. Our analysis revealed a significant decrease in the relative telomere length in old wildtype, young mutant, and old mutant animals (Figure 9b), thereby supporting the roles of Grna and Grnb in regulating physiological aging kinetics in the adult zebrafish brain.

## 4. Discussion

Despite numerous studies performed in the context of neurodegeneration and wound healing that have revealed the association of GRN with various processes including wound healing, inflammation, lysosome regulation, neuronal survival, and neurodegeneration [86,87], the physiological roles of GRN in the central nervous system (CNS) remain elusive. Here, we identified Grna and Grnb as key regulators of the aging kinetics in the adult zebrafish telencephalon. Specifically, Grna and Grnb deficiency in microglial cells led to a pro-inflammatory phenotype, which has been previously associated with aging [88]. Moreover, we observed additional aging-associated changes including a decrease in constitutive neurogenesis and oligodendrogenesis, and telomere shortening. The phenotype of changes observed in Grna; Grnb-deficient animals was also observed in wildtype animals during aging, thus supporting the role of Grna and Grnb in regulating aging kinetics in the zebrafish telencephalon. Our data are in line with the association of the *GRN* gene locus with aging in the human cerebral cortex [33] as well as with evidences that GRN suppresses inflammation in the mouse brain during aging [38]. Interestingly, the levels of Grnb physiologically decreased during aging in the adult zebrafish telencephalon, thus partially mimicking the levels observed in mutant animals. These results suggest that progressive loss of Grnb may be a key determinant of physiological aging in the adult zebrafish telencephalon. 

Our data show that Grna and Grnb deficiency induces a pro-inflammatory phenotype in microglial cells, which display larger cell bodies, hyper-ramifications and shorter processes- morphological characteristics described in microglial cells during aging in the mouse brain [89]. The altered morphology of mutant microglial cells was accompanied by an increase in the levels of numerous pro-inflammatory cytokines and complement factors as well as a decrease in the levels of anti-inflammatory cytokines. Indeed, one of the most prominent features of brain aging is the activated state of immune cells, thus leading to the secretion of pro-inflammatory cytokines that might not only affect neurons and glia in the central nervous system, but also induce systemic aging-related phenotypes [90]. However, how microglial cells are activated in the brain in Grna; Grnb-deficient animals remains elusive. Comparing the transcriptomes of wildtype and mutant microglial cells, we detected significant upregulation of numerous genes associated with MAPK and JAK/STAT signalling pathways, which are known to be involved in the regulation of pro-inflammatory cytokine production and microglial activation [66,67]. Similar results have been observed in the mouse brain, where GRN has been identified as a key regulator of inflammation, by tuning the expression of pro-inflammatory cytokines and complement factors during aging [38], thus suggesting that granulins directly adjust the levels of inflammatory cytokines to the corresponding aging state in both zebrafish and mammalian brains.

Because we hypothesised that Grna and Grnb directly regulate the kinetics of aging in the adult zebrafish telencephalon and speculated that Grna; Grnb deficiency cause a premature aging phenotype in the telencephalon in young animals, we examined the transcriptional signature in the entire telencephalon in wildtype and mutant animals at two different ages (young and old animals). Indeed, aging caused significant changes in the transcription levels of numerous genes in the telencephalon in old wildtype animals. Unexpectedly, a high proportion (~50%) of DEGs in old wildtype animals were also altered in same trend as that in young mutant animals, thereby indicating a similar transcriptome signature. The set of common DEGs was enriched in programs associated with innate immunity, regulation of apoptotic process, extracellular matrix composition, and p53 signalling; similar enrichment has also been identified in the transcriptome of *Notobranchius furzeri*, a well-established model of brain aging [72], thus supporting the concept of a premature aging phenotype in the telencephalon in young mutant animals. Importantly, a large proportion of common DEGs included genes that were not regulated in mutant microglial cells, thus suggesting that granulin deficiency not only affects microglia but also induces additional aging-related changes in other cell populations. However, whether these effects are due to a direct function of Grna and Grnb in these cell types or whether the observed changes are indirect and induced by altered mutant microglial cells remains unclear. 

Interestingly, we identified diminished mRNA levels of *il4* and *stat6* in the telencephalon in old wildtype, young mutant, and old mutant animals. Il4 and Stat6 have been implicated in the regulation of neuronal progenitor proliferation and neurogenesis [75], whose rates are directly controlled by microglial activity [91,92,93]. Therefore, our data, at least in part, support the hypothesis that the proinflammatory phenotype of Grna; Grnb-deficient microglial cells contributes to the aging of the zebrafish neural stem cell compartment. Moreover, the ependymoglial population (the neural stem cells in the adult zebrafish telencephalon) displays an age-related increased proportion of quiescent cells, thereby decreasing the number of newly generated neurons [77]. Indeed, we demonstrated that both aging and Grna; Grnb deficiency in young animals caused a similar reduction in the number of proliferating cells (mostly neuronal progenitors in the neural stem cell niche) and newly generated neurons, thus strengthening the association between granulins and regulation of brain aging. Moreover, both aging and Grna; Grnb deficiency in young animals led to a decrease in oligodendrogenesis, a process severely affected in the aging CNS in both mice and zebrafish [77,94,95]. Furthermore, the decreased oligodendrogenesis could be explained by dramatic changes in the transcriptome of mutant oligodendroglial cells. In fact, GO term analysis revealed downregulation of several genes associated with Hedgehog, canonical Wnt, Notch, TGF-β, ErbB signalling, and with negative regulation of BMP signalling pathways. Notably, all these pathways have been associated with oligodendroglial cell maturation, differentiation and myelination [80,81,82,83,84], and they depend on niche-provided signals amenable to changes by microglial cells [96]. These findings further support the idea that initial changes in microglial cells trigger age-related changes in neurogenesis and oligodendrogenesis in a cell non-autonomous manner.

Finally, we also observed significant telomere shortening in the brain in young mutant animals, to levels comparable to those detected in aging brains. Moreover, we detected diminished mRNA levels of genes encoding for enzymes involved in telomere length maintenance. Because telomere shortening is a typical cell-intrinsic features associated with aging [97], Grna and Grnb might coordinate the acquisition of both cell-extrinsic and -intrinsic aging features in the adult zebrafish brain.

In this study, we identified the roles of Grna and Grnb in regulating the inflammatory state under physiological conditions and during aging in the zebrafish brain, in line with observations in the mouse CNS [38,98]. Furthermore, we provide new evidence that Grna and Grnb directly regulate aging and that their loss causes a premature aging phenotype in the telencephalon in young zebrafish, thus providing new insights into the physiological roles of granulins that could potentially be used for the development of new therapeutic interventions in the mammalian CNS.

## Figures and Tables

**Figure 1 cells-09-00350-f001:**
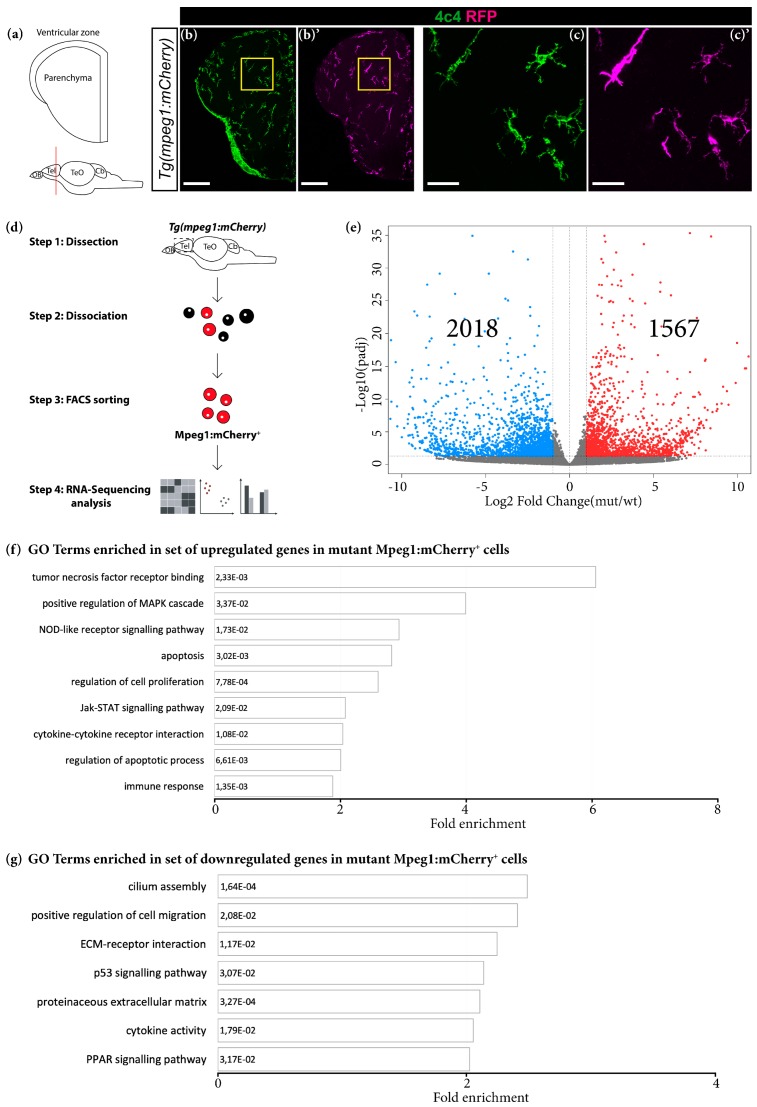
Grna and Grnb deficiency leads to transcriptional changes associated with an activated microglial state. (**a**) Schematic of a typical section from the adult zebrafish telencephalon. (**b**–**c′**) Micrographs depicting microglial cells in the telencephalon in *Tg* (*mpeg1:mCherry*) animals. (**c**,**c′**) Magnifications of areas boxed in (**b**) and (**b′**), respectively. Scale bars: 100 µm (**b**,**b′**) and 20 µm (**c**,**c′**). (**d**) Scheme depicting the isolation procedure of Mpeg1^+^ cells for transcriptome analysis. (**e**) Volcano plot of differentially expressed genes (DEGs) in mutant Mpeg1^+^ cells (*padj* ≤ 0.05; 1 ≤ log2FC ≤ −1). (**f**,**g**) Histograms depicting gene ontology (GO) terms enriched in the set of upregulated (**f**) and downregulated (**g**) genes in mutant Mpeg1^+^ cells.

**Figure 2 cells-09-00350-f002:**
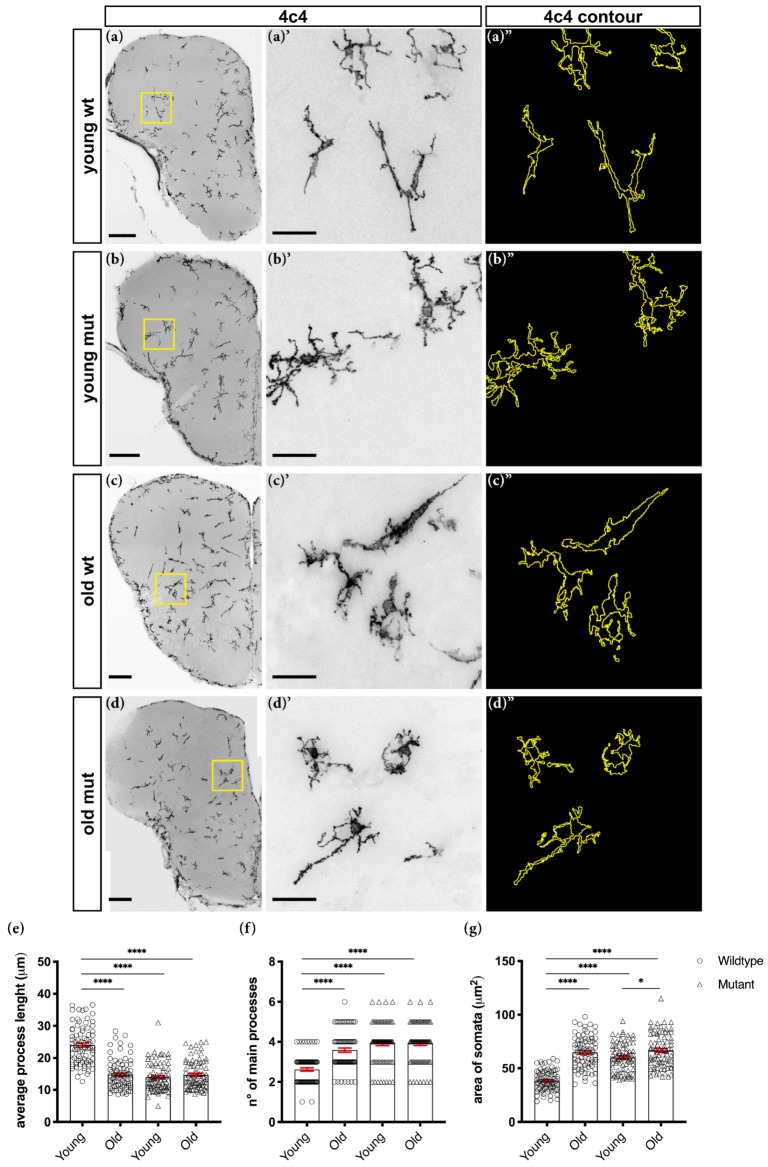
Grna; Grnb-deficient microglial cells display morphological changes associated with aging. (**a**–**d**) Micrographs depicting 4C4^+^ microglial cells. (**a′**–**d′**) Magnifications of boxed areas in (**a**–**d**), respectively. (**a″**–**d″**) Morphological tracing of 4C4^+^ microglial cells. Scale bars: 100 µm (**a**–**d** magnification) and 20 µm (**a′**–**d′**). (**e**–**g**) Morphological analysis depicting the average process length (**e**), number of main processes (**f**) and area of somata (**g**) of 4C4^+^ microglial cells. n.s, not significant, * *p* ≤ 0.05, **** *p* ≤ 0.0001**,** each data point represents a single cell (a total of 80 cells from four telencephali were analysed in each group).

**Figure 3 cells-09-00350-f003:**
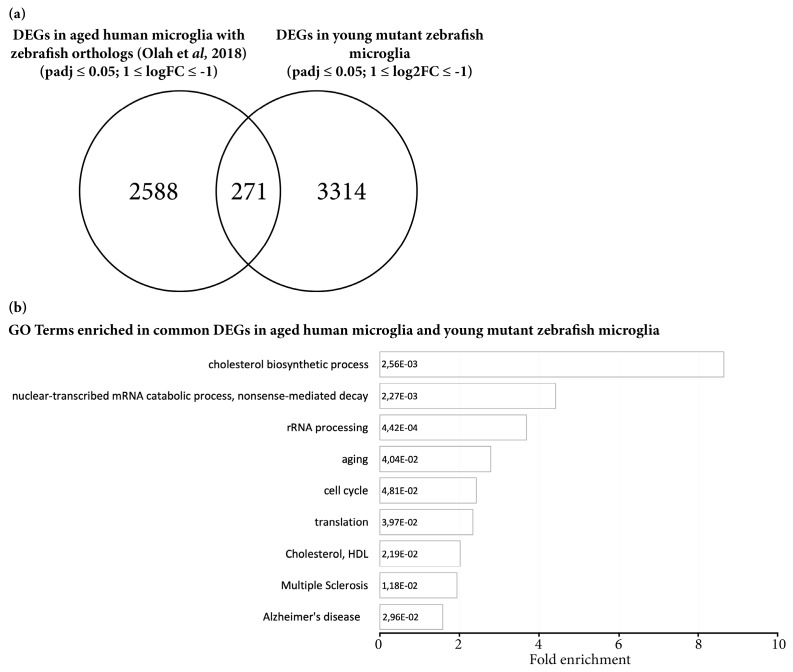
Microglial cells isolated from young mutant zebrafish share a proportion of DEGs with aging human microglial cells. (**a**) Venn diagram depicting common DEGs in aging human microglial cells with existing zebrafish orthologs and in young mutant zebrafish Mpeg1^+^ cells. (**b**) Histograms depicting gene ontology (GO) terms enriched in the set of 271 common DEGs in aging human and young mutant zebrafish microglial cells.

**Figure 4 cells-09-00350-f004:**
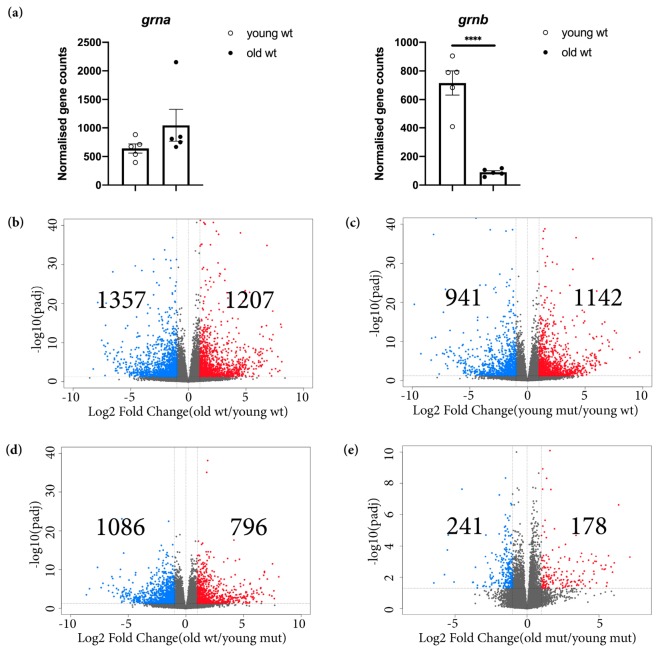
Transcriptomic changes detected in old wildtype, young mutant, and old mutant telencephali. (**a**) Normalised gene counts of *grna* and *grnb* in young and old wildtype animals. **** *padj* ≤ 0.0001, each data point represents a distinct biological replicate (five telencephali per group were analysed). (**b**) Volcano plot of DEGs in old wildtype animals (young wildtype animals were used as a reference) (*padj* ≤ 0.05; 1 ≤ log2FC ≤ −1). (**c**) Volcano plot of DEGs in young mutant animals (young wildtype animals were used as a reference) (*padj* ≤ 0.05; 1 ≤ log2FC ≤ −1). (**d**) Volcano plot of DEGs in old wildtype animals (young mutant animals were used as a reference) (*padj* ≤ 0.05; 1 ≤ log2FC ≤ −1). (**e**) Volcano plot of DEGs in old mutant animals (young mutant animals were used as a reference) (*padj* ≤ 0.05; 1 ≤ log2FC ≤ −1).

**Figure 5 cells-09-00350-f005:**
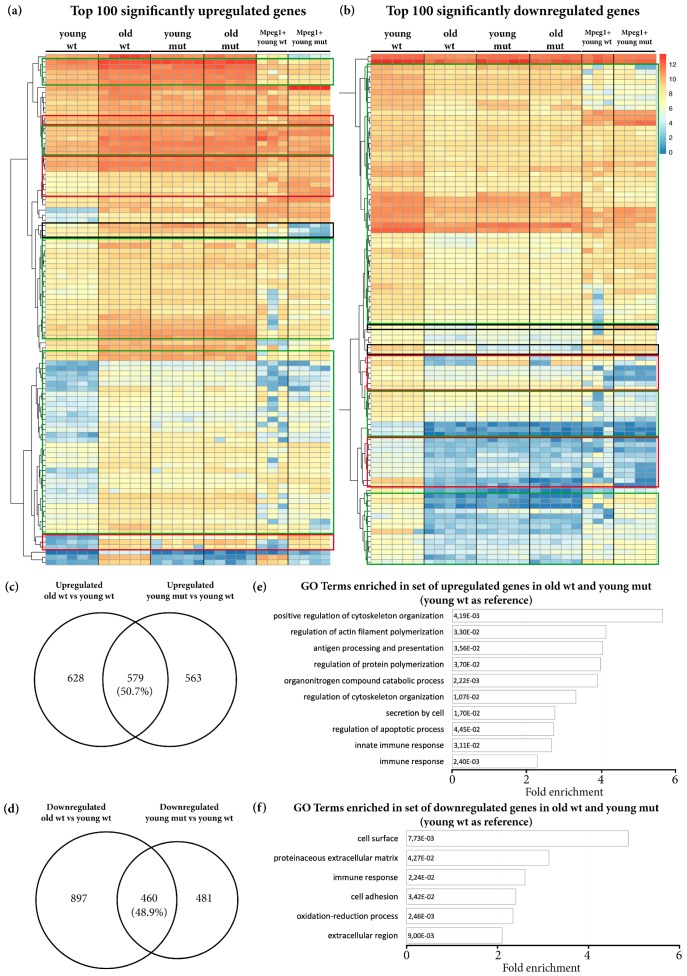
The phenotype of age-related transcriptional changes detected in old wildtype animals is partially mimicked by Grna and Grnb deficiency in young mutant animals. (**a**,**b**) Heat maps depicting rlog-transformed values of the top 100 significantly upregulated and downregulated genes in old wildtype telencephali (young wildtype were used as a reference) in young wildtype animals, old wildtype animals, young mutant animals, old mutant animals, young wildtype Mpeg1^+^ cells, and young mutant Mpeg1^+^ cells. Green boxes: DEGs in old wildtype animals with the same direction of regulation in young mutant and old mutant animals, but not regulated in mutant microglial Mpeg1^+^ cells. Red boxes: DEGs in old wildtype animals with comparable levels in young mutant and old mutant animals, regulated in the same direction in mutant microglial Mpeg1^+^ cells. Black boxes: DEGs in old wildtype animals with comparable values in young mutant and old mutant animals, regulated in the opposite direction in mutant microglial Mpeg1^+^ cells. (**c**,**d**) Venn diagrams depicting common DEGs in old wildtype and young mutant telencephali (young wildtype were used as a reference). (**e**,**f**) Histograms depicting gene ontology (GO) terms enriched in the set of common upregulated and downregulated genes in old wildtype and young mutant telencephali (young wildtype were used as a reference).

**Figure 6 cells-09-00350-f006:**
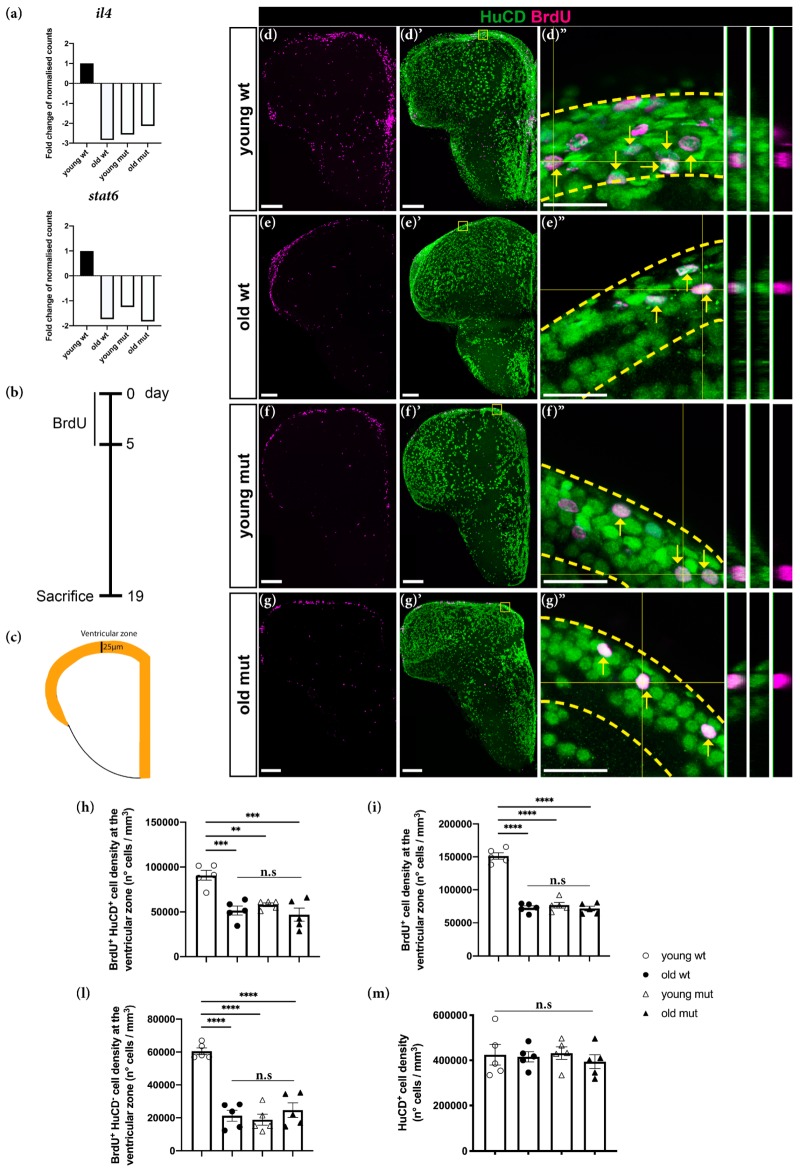
Grna and Grnb deficiency mimics the phenotype of the decrease in neurogenesis observed during aging in the adult zebrafish telencephalon. (**a**) Histograms depicting relative expression (normalised to that in young wildtype animals) of *il4* and *stat6*. (**b**) Experimental scheme to assess neurogenesis. (**c**) Schematic of a typical section from the adult zebrafish telencephalon. Orange area: ventricular area analysed in the experiment. (**d**–**g″**) Micrographs depicting newly generated neurons (arrow). (**d″**–**g″**) Magnifications with orthogonal projections of boxed areas in (**d′**–**g′**), respectively. Scale bars: 100 µm (**d**–**g′**) and 20 µm (**d″**–**g″**). (**h**–**l**) Histograms depicting the density of newly generated neurons (**h**), BrdU^+^ cells (**i**), and density of non-neuronal BrdU-label retaining cells (**l**) in the ventricular zone of the adult zebrafish telencephalon. (**m**) Histogram depicting the density of total HuCD^+^ neurons. n.s, not significant, ** *p* ≤ 0.01, *** *p* ≤ 0.001, **** *p* ≤ 0.0001, each data point represents a distinct biological replicate (five telencephali per group were analysed).

**Figure 7 cells-09-00350-f007:**
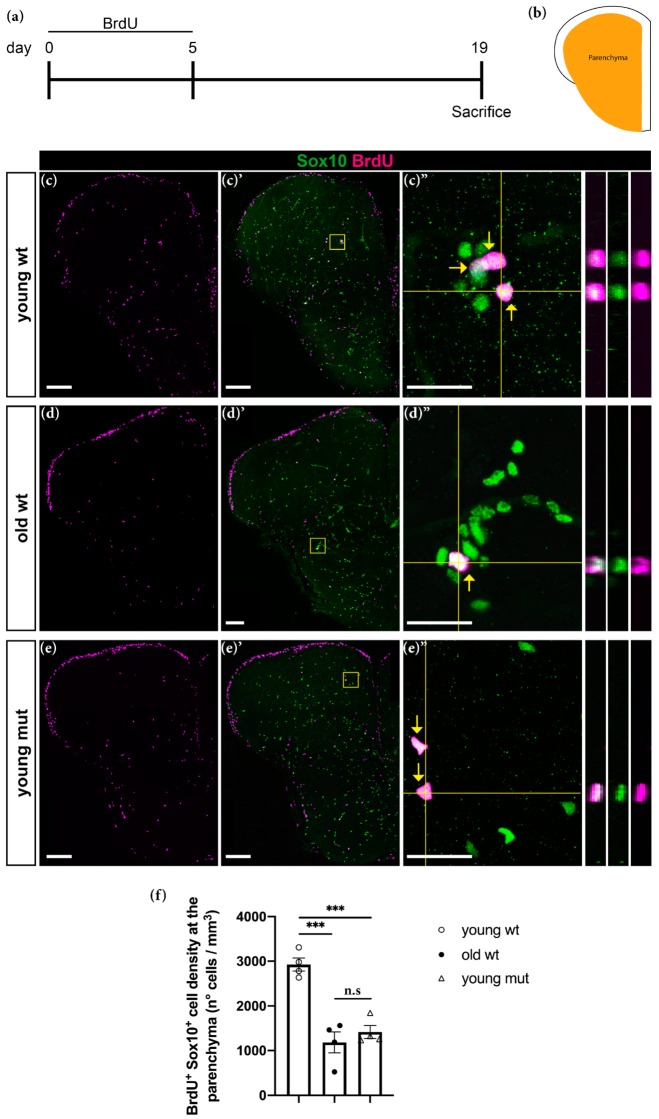
Grna; Grnb deficiency mimics the phenotype of the decrease in oligodendrogenesis observed during aging in the adult zebrafish telencephalon. (**a**) Experimental scheme. (**b**) Schematic of a typical section from the adult zebrafish telencephalon. Orange area: parenchymal area analysed in the experiment. (**c**–**e″**) Micrographs depicting BrdU positive oligodendroglial cells stained with Sox10 (green). (**c″**–**e″**) Magnifications with orthogonal projections of boxed areas in (**c′**–**e′**), respectively. Yellow arrows depict cells positive for Sox10 and BrdU. Scale bars: 100 µm in (**c**–**e′**) and 20 µm in (**c″**–**e″**). (**f**) Histogram depicting the density of BrdU^+^ Sox10^+^ cells in the parenchyma. n.s, not significant, *** *p* ≤ 0.001, each data point represents a distinct biological replicate (four telencephali per group were analysed).

**Figure 8 cells-09-00350-f008:**
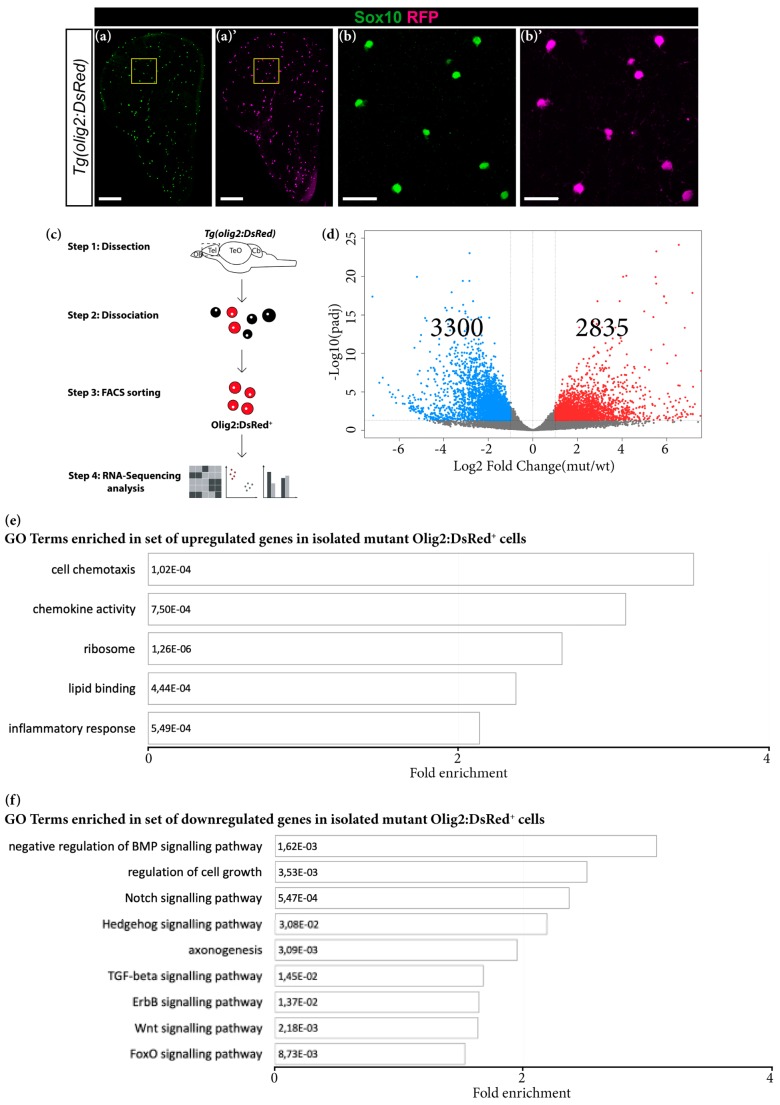
Transcriptomic changes in FACS-isolated mutant oligodendroglial cells. (**a**–**b′**) Micrographs showing perfect co-localisation of DsRed in the *Tg* (*olig2:DsRed*) line with the oligodendrocyte lineage marker Sox10. (**b**,**b′**) Magnifications of the boxed areas in (**a**) and (**a′**), respectively. Scale bars: 100 µm in (**a**,**a′**) and 20 µm in (**b**,**b′**). (**c**) Scheme depicting isolation procedure of Olig2^+^ cells for transcriptome analysis. (**d**) Volcano plot of differentially regulated genes in mutant Olig2^+^ cells (*padj* ≤ 0.05; 1 ≤ log2FC ≤ −1). (**e**,**f**) Histograms depicting gene ontology (GO) terms enriched in the set of upregulated (**e**) and downregulated (**f**) genes in mutant Olig2^+^ cells.

**Figure 9 cells-09-00350-f009:**
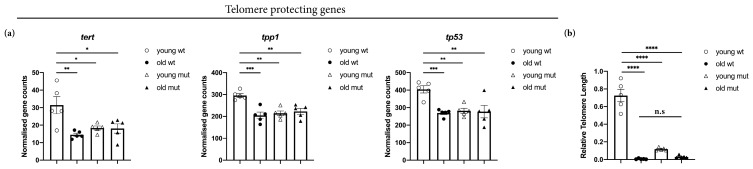
Grna and Grnb deficiency causes downregulation of telomere-protective genes and telomere shortening in the telencephalon in young zebrafish, as observed in old animals. (**a**) Histogram depicting normalised gene counts of *tert*, *tp53*, and *tpp1*. (**b**) Histogram depicting relative telomere length in the young wildtype, old wildtype, young mutant, and old mutant zebrafish telencephalon. *n.s* not significant, * *padj* ≤ 0.05, ** *padj* ≤ 0.01, *** *padj* ≤ 0.001, **** *padj* ≤ 0.0001, each data point represents a distinct biological replicate (five telencephali per group were analysed).

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
