# Peer review of "Granulins Regulate Aging Kinetics in the Adult Zebrafish Telencephalon"

_cells, 2020, doi:10.3390/cells9020350_

Round 1
Reviewer 1 Report
The presented work is devoted to a interesting and little-studied problem associated with the participation of Granilis in the processess of brain aging. The authors presented an interesting and comprehensive experimental data on the Danio telencephalon to clarify the affects of granulins on inflamatory phenotype of microglia and brain aging. In general, the work is commendable, but in the review process a number significant comments and quastions arose:
1) Research methods are described very poorly. There is no clarity in the methodologic aspects, in particular, it is not clear what the total number of animals was used in this work. The experimenral design is poorly described, the scheme of BrdU administration and the proliferative activity of glial cells are not clear. There is no explanation why, it was precisely the indicated timing of BrdU exposure that was chosen to detect nuerogenesis and oligodendrogenesis. It is not clear why 5 telencephalic per groups were used to assess proliferation and constitutive neurogenesis in the ventricular zone, and 4 per group were used to analyze oligodendrogenesis in the telencephalic parenchyma. From the presentes in Fug. 6B and 7B, it is not clear how cells density was determined and cells volume were calculated. It is also not clear how the normalisation of cells volumes was carried out. Genetic methods of analysis are also poorly described and need to be processed and detailed explanations.
2) The Result section contains a lot of abbreviations, which greatly copmlicates the understanfing of the main text and meaning of the results; the list of abbreviations is necessary. The Results section often contained the literature references: page 8, 9 et al., in the manuscript. Supplemenraty files include 10 tables, containing very large quatitative material. Tables are poorly analysed. It is necessary to analyze the material in supl. tables and in the Results section give generelized most important data from these tables.
3) The Result section needs in grammatical correction p. 17 last paragraph
4) In the Discussion section, the results are often repeated.
Author Response
Response to editorial comments
According to the guidelines, we carefully checked that citations within the text and the references at the end of the text are in the correct format.
We updated the figures in the text at appropriate positions in high quality.
We also added new graphs to Figure 2 and Figure 6 and provided new Supplementary Figures.
The manuscript was edited by native English speaker and the certificate is now provided.
All the revisions were clearly highlighted using “Track Changes” function in Microsoft Word.
Detailed responses to Reviewer's comments
Reviewer 1:
The presented work is devoted to a interesting and little-studied problem associated with the participation of Granilis in the processess of brain aging. The authors presented an interesting and comprehensive experimental data on the Danio telencephalon to clarify the affects of granulins on inflamatory phenotype of microglia and brain aging. In general, the work is commendable, but in the review process a number significant comments and quastions arose:
1) Research methods are described very poorly. There is no clarity in the methodologic aspects, in particular, it is not clear what the total number of animals was used in this work.
We have now edited the material and method section to make it more structured and to provide an additional information. Moreover, “Quantitative analysis and statistical tests” paragraph in Material and Methods was updated to clarify the number of animals that we used in different experiments. Additionally, all figure legends were updated, clearly specifying the number of animals analysed in each experiment (Page 5, lines 132-135).
2) The experimenral design is poorly described, the scheme of BrdU administration and the proliferative activity of glial cells are not clear. There is no explanation why, it was precisely the indicated timing of BrdU exposure that was chosen to detect nuerogenesis and oligodendrogenesis.
“BrdU labelling experiments” paragraph in Material and Methods was updated to provide explanations for the timing of BrdU exposure based on previous findings.
We think that the schemes, together with the updated paragraph, are sufficiently explanatory to understand the experimental design (Page 4, lines 118-126).
3) It is not clear why 5 telencephalic per groups were used to assess proliferation and constitutive neurogenesis in the ventricular zone, and 4 per group were used to analyze oligodendrogenesis in the telencephalic parenchyma.
We have performed statistical analysis prior to experiments that is based on the variations of the measured parameters in our previous research and determined that we need the group of minimum 4 animals to achieve the statistical confidence at 95%. To make sure that we reach group size of 4 and at the same time reduce use of animals to the minimum, we aimed at total 5 animals per experimental group. In one experiment aiming to analyse the oligodendrogenesis in the telencephalic parenchyma, one animal in one group died during experiment. To have equal statistical power among groups, we decided to keep four animals in each group for oligodendrogenesis experiment.
4) From the presentes in Fug. 6B and 7B, it is not clear how cells density was determined and cells volume were calculated. It is also not clear how the normalisation of cells volumes was carried out.
“Quantitative analysis and statistical tests” paragraph in Material and Methods was updated to clarify how cell density and volumes were calculated. Additionally, we better clarified the normalisation procedure (Page 5, lines 136-141). We thank the Reviewer for the valid point that will help readers.
5) Genetic methods of analysis are also poorly described and need to be processed and detailed explanations.
“DNA extraction and genotyping” paragraph in Material and Methods was updated to describe how genetic variants of grna and grnb were analysed (Page 4, lines 100-105).
6) The Result section contains a lot of abbreviations, which greatly complicates the understanfing of the main text and meaning of the results; the list of abbreviations is necessary.
We agree with the fact that numerous abbreviations would greatly complicate the general understanding of the main text. For this reason, as suggested by the Reviewer, we included a list of abbreviations.
7) The Results section often contained the literature references: page 8, 9 et al., in the manuscript.
We acknowledge the Reviewer´s point. However, we also think that references in Results simplify the general understanding of the text and strengthen the main concepts that we want to provide to the readers. For this reason, we would like to keep the references in the Results section.
8) Supplemenraty files include 10 tables, containing very large quatitative material. Tables are poorly analysed. It is necessary to analyze the material in supl. tables and in the Results section give generelized most important data from these tables.
Thanks to the Reviewer´s comment, we realised that the references for the Supplementary tables provided were not updated in the Results section. We now updated references to the Supplementary tables to support our statements about general pathways and processes affected by either aging or mutation.
Additionally, we provided heat maps containing regulated genes enriched in the most meaningful GO terms described in the text, as additional guidelines for the reader (Figure S3, S5, S6, S8).
9) The Result section needs in grammatical correction p. 17 last paragraph
The manuscript got revised by native English editor and the certificate is attached to the revised version of manuscript.
10) In the Discussion section, the results are often repeated.
Our concept was that we shortly mention results and then discuss them in depth. We think that this helps reader integrating data and would like to keep such organization.

Reviewer 2 Report
In this manuscript, Zambusi et al. sought to examine the molecular mechanisms underlying age-associated phenotypes in zebrafish with Progranulin (Grn) deficiency. The authors performed numerous transcriptome analyses on brain tissues and FACS-isolated microglia/oligodendrocytes from Grn-deficient and control animals, at two different ages. Importantly, compared to age-match controls, Grn deficiency in young animals altered gene expression, and many of these gene expression changes were observed during physiological aging. In addition, they further demonstrated that depletion of Grn causes microglial activation, impaired neurogenesis/oligodendrogenesis, and telomere shortening, indicating roles for Grn in brain aging. Indeed, potential contributions of Grn to age-associated impairments and gene expression changes have been a topic of interest in the field, but have not been described in a zebrafish model. I feel that the manuscript is a strong candidate for publication in Cells, but have a number of comments that if addressed should help to clarify the findings and improve the manuscript.
Major comments:
Lysosomal dysfunction has been demonstrated in mice lacking GRN, and transcriptome analyses have also supported the relationship between GRN and lysosomal function. However, no such enriched GO term was observed throughout this manuscript. Could the authors comment on this discrepancy? Moreover, the authors should compare their differentially expressed genes to the published dataset (Lui et al., Cell, 2016), in which multiple brain regions and FACS-isolated microglia from Grn knockout mice and age-match controls were used for RNA-seq analysis. Several concepts are not explained. For instance, is Grn expressed ubiquitously in main brain cell types? Are there functional deficits observed in Grn-deficient animals? Does the cell type composition alter (e.g. gliosis or neuronal loss) in Grn-deficient animals? Including this information in the text would help the readers to better understand the rationales and the purposes of each experiment. Quantification of morphological changes in microglia should be performed to support the authors’ statement in Figure 2. Measuring process length of microglia using ImageJ software should be very straightforward. To keep consistency of this manuscript, the old mutant group should be assessed in results presented in Figure 7 and Figure 9b. Increase the font size of Figure 1a, 1c, and 8c would be helpful.
Author Response
Response to editorial comments
According to the guidelines, we carefully checked that citations within the text and the references at the end of the text are in the correct format.
We updated the figures in the text at appropriate positions in high quality.
We also added new graphs to Figure 2 and Figure 6 and provided new Supplementary Figures.
The manuscript was edited by native English speaker and the certificate is now provided.
All the revisions were clearly highlighted using “Track Changes” function in Microsoft Word.
Detailed responses to Reviewer's comments
Reviewer 2:
In this manuscript, Zambusi et al. sought to examine the molecular mechanisms underlying age-associated phenotypes in zebrafish with Progranulin (Grn) deficiency. The authors performed numerous transcriptome analyses on brain tissues and FACS-isolated microglia/oligodendrocytes from Grn-deficient and control animals, at two different ages. Importantly, compared to age-match controls, Grn deficiency in young animals altered gene expression, and many of these gene expression changes were observed during physiological aging. In addition, they further demonstrated that depletion of Grn causes microglial activation, impaired neurogenesis/oligodendrogenesis, and telomere shortening, indicating roles for Grn in brain aging. Indeed, potential contributions of Grn to age-associated impairments and gene expression changes have been a topic of interest in the field, but have not been described in a zebrafish model. I feel that the manuscript is a strong candidate for publication in Cells, but have a number of comments that if addressed should help to clarify the findings and improve the manuscript.
1) Lysosomal dysfunction has been demonstrated in mice lacking GRN, and transcriptome analyses have also supported the relationship between GRN and lysosomal function. However, no such enriched GO term was observed throughout this manuscript. Could the authors comment on this discrepancy?
We thank the Reviewer for the valid point as lysosomal disfunction is an important phenotype related to GRN deficiency. For this reason, we added a comparison of regulated lysosomal genes previously identified in GRN-deficient mice (Evers et al, 2017 and Götzl et al, 2018). In this regard, we compared and provided the normalised gene counts for lysosomal genes in young/old and wildtype/mutant animals in Figure S7. Additionally, we added a paragraph (Page 13, lines 394-402) to describe the results obtained. Differently from what observed in the mouse brain, no major differences on lysosomal gene expression were detected in mutant zebrafish. This discrepancy between mouse and zebrafish is now discussed in the manuscript.
2) Moreover, the authors should compare their differentially expressed genes to the published dataset (Lui et al., Cell, 2016), in which multiple brain regions and FACS-isolated microglia from Grn knockout mice and age-match controls were used for RNA-seq analysis.
We thank the Reviewer for the suggestion. The published dataset in Lui et al., Cell, 2016 contains the age-dependent upregulated genes in the cortex of GRN-deficient mice enriched in two major categories (lysosome and immunity). In the publication, it is stated that all the regulated genes identified are associated with microglial cells. For this reason, we compared the regulated genes from the main categories in Lui et al. Cell, 2016, with the regulated genes in mutant Mpeg1+ microglial cells. The results of this comparison were added to Table S1, together with an explanatory paragraph in the Result section (Page 8, lines 251-262).
3) Several concepts are not explained. For instance, is Grn expressed ubiquitously in main brain cell types? Are there functional deficits observed in Grn-deficient animals? Does the cell type composition alter (e.g. gliosis or neuronal loss) in Grn-deficient animals? Including this information in the text would help the readers to better understand the rationales and the purposes of each experiment.
We agree with the Reviewer that the following information would help the readers to better understand the rationales of experiments.
References about Grn expression in mouse were added to Introduction section.
grna and grnb expression levels in main cell types were added to Figure S2. Unfortunately, expression levels of grna and grnb in isolated neurons are not available.
Neuronal loss was not identified between groups based on quantification of HuCD+ cell density (added to Figure 6) (Page 15, lines 472-475). This result could explain the absence of obvious functional deficits detected in mutant zebrafish.
4) Quantification of morphological changes in microglia should be performed to support the authors’ statement in Figure 2. Measuring process length of microglia using ImageJ software should be very straightforward.
We agree with Reviewer´s comment. For this reason, quantifications of average process length, number of main ramifications and area of somata were added to Figure 2. Short explanatory paragraph was added to Materials and Methods section (Page 5, lines 150-153).
5) To keep consistency of this manuscript, the old mutant group should be assessed in results presented in Figure 7 and Figure 9b.
Results for old mutant group could be added for telomere experiment readily as we had brains previously stored at -80°C. These experiments are now included in Figure 9b.
Regarding the oligodendrogenesis experiment (Figure 7), unfortunately we do not have mutant animals of the proper age. Adding this experimental group would require additional 12 months that is beyond the revision time. However, since all other results would support the concept that no major differences exist between young mutant and old mutant animals, this information, in our view, would not bring any further conceptual advances.
6) Increase the font size of Figure 1a, 1c, and 8c would be helpful.
Font was increased in the abovementioned figures.
Round 2
Reviewer 1 Report
The revised version of the article significantly improved the content, compared to the original version of the manuscript; most of the remarks related to clarifying certain issues were corrected satisfactorily. However, a large number of supplementary files (18!) still make it difficult to understand the content and evaluate the results of article.
Reviewer 2 Report
In the revised manuscript, the authors have provided new evidence and additional analyses in response to several of the concerns raised previously. The manuscript has been strengthened and I can now recommend the revised version for publication in Cells.
